# Lasing of a cavity-based X-ray source

Patrick Rauer[1✉], Immo Bahns[2,4], Bertram Friedrich[2], Sara Casalbuoni[2], Massimiliano Di Felice[2], Martin Dommach[2], Idoia Freijo Martin[2], Wolfgang Freund[2], Jan Grünert[2], Marc Guetg[1], Ivars Karpics[2], Suren Karabekyan[2], Andreas Koch[2], Naresh Kujala[2], Daniele La Civita[2], Jia Liu[2], Theophilos Maltezopoulos[2], Mikako Makita[2], Frank Mayet[1], Lukas Müller[1], Benoit Rio[2], Liubov Samoylova[2], Silja Schmidtchen[2], Matthias Scholz[1], Alessandro Silenzi[2], Vivienne Strauch[2], Daniel Thoden[1], Torsten Wohlenberg[1], Maurizio Vannoni[2], Fan Yang[2], Winfried Decking[1], Joerg Rossbach[3] & Harald Sinn[2]

The invention of the laser transformed optics by providing intense, coherent light in the visible region, but extending this concept to X-rays has been hindered by a lack of suitable gain media and mirrors. Current hard X-ray free-electron laser (XFEL) facilities[1–5] overcome this by amplifying shot noise from a high-peak-current electron bunch via self-amplified spontaneous emission[6] in a single pass through long undulators, delivering very high brightness but with a noisy, multi-spiked temporal and spectral profile. Cavity-based XFELs (CBXFELs)[7–9] were proposed to close this gap by recirculating spectrally filtered X-ray pulses in a Bragg-reflecting cavity synchronized to a high-repetition-rate electron beam. Here we show lasing with multi-pass gain at 6.952 keV in a 132.8-m round-trip diamond-based Bragg cavity[10] at the European XFEL, matched to the 2.23-MHz bunch spacing of the superconducting accelerator[5]. Under stringent length and angular stability requirements, a ring-up in the cavity across successive bunches was observed, producing spectrally pure, microjoule-level pulses. This establishes the feasibility of CBXFELs in an accelerator environment and validates diamond Bragg optics for X-ray resonators. The demonstrated spectral purity opens a path to next-generation X-ray science, which demands highly coherent, stable sources.

Shortly after the free-electron laser (FEL) concept was proposed in the 1970s[11], the first FEL oscillator was developed[12] and many such systems were soon in operation worldwide in the infrared to ultraviolet wavelength regime[13], where high-reflectivity mirrors were well established. Lacking suitable cavity optics, hard X-ray FELs adopted single-pass, high-gain self-amplified spontaneous emission (SASE)[6]. However, SASE radiation shows significant shot-to-shot fluctuations and is limited in longitudinal coherence compared with visible-light lasers, with typical bandwidths of ≥0.1% (ref. 14). A significant improvement has been achieved with the self-seeding technique[15–19], where a crystal in Bragg reflection is inserted after an initial undulator section and the diffracted, filtered radiation is then used to seed a new lasing process in the downstream part of the undulator. This improves spectral purity and spectral flux, yet it does not fully address stability and coherence limitations. An X-ray laser oscillator was proposed[20] that exploits stimulated emission from atomic inner-shell transitions[21–23] within an X-ray optical cavity pumped by an X-ray free-electron laser (XFEL) beam, but a complete X-ray laser oscillator has not yet been realized.

As an alternative approach, cavity-based XFELs (CBXFELs)[7–9], including both high-gain regenerative X-ray amplifier FELs[7] and low-gain XFEL oscillators[8], trap and monochromatize X-ray pulses in a Bragg-reflecting cavity around only a few undulator segments and synchronize them to high-repetition-rate electron bunches. This oscillator scheme has the potential to generate fully coherent, highly stable and spectrally

pure X-ray pulses analogous to the infrared/ultraviolet oscillators, while reducing the requirement for very long undulator lines[10,24–34].

Realizing such a device requires accelerators with sufficiently high peak brightness, capable of delivering at least tens of electron bunches at a repetition rate high enough to confine the cavity length to a practical scale of several tens of metres[5,35]. Owing to their exceptional thermal characteristics and high reflectivity, diamond single crystals are uniquely suited as Bragg reflectors for this application[36]. Still, there are considerable technical challenges to overcome to realize such a cavity in an accelerator environment, as it requires stability and alignment tolerances at the nanoradian level, while adjusting the overall length of the cavity to micrometre accuracy[37].

Recently, low-loss ring-down of X-rays in a diamond-based cavity without undulators was demonstrated at the Linac Coherent Light Source[38], showing the feasibility of the X-ray optical requirements, but no FEL amplification.

Here we report on the demonstration of lasing in an X-ray cavity with intra-cavity undulators[10]. A schematic of the experimental set-up is shown in Fig. 1. The last 4 segments with 20-m active length of the European XFEL SASE1 undulator line were used at an electron energy of 14 GeV. The undulator segments are enclosed by a pair of retro-reflecting units at a distance of 66.41 m (cavity path length 132.8 m), each consisting of 1 diamond crystal and 2 perpendicular X-ray mirrors. The cavity length was fine-tuned to the electron bunch

[1]Deutsches Elektronen-Synchrotron DESY, Hamburg, Germany. [2]European XFEL GmbH, Schenefeld, Germany. [3]Universität Hamburg, Hamburg, Germany. [4]Present address: Byonoy GmbH, Hamburg, Germany. ✉e-mail: patrick.rauer@desy.de

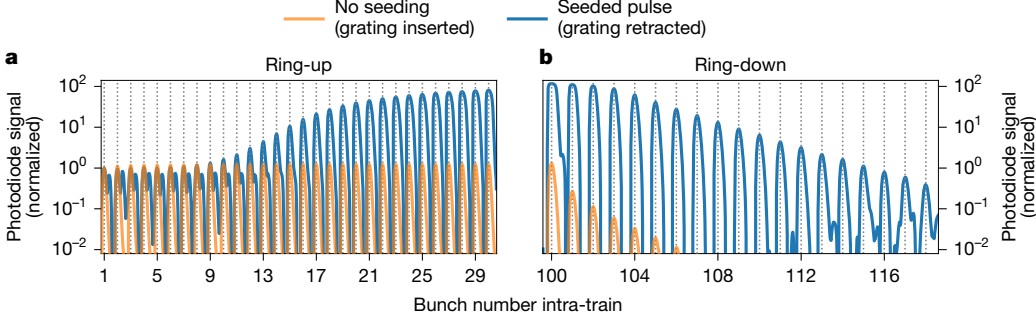

**Fig. 1 | Experimental set-up.** Schematic of the CBXFEL experiment at the SASE1 undulator line of the European XFEL. The X-ray cavity consists of two diamond crystals in Bragg reflection working close to backscattering geometry and two pairs of focusing X-ray mirrors to steer and focus the beam. The surfaces of the crystal and the mirrors are perpendicular to each other, forming a retro-reflector. The elements in the retro-reflector are aligned such that the beam after reflection returns at a distance of 2.67 mm from the incident beam inside the same vacuum pipe. The downstream retro-reflector unit (on the right) can be translated to tune the cavity length. The electron beam (blue) is injected from the left using a magnetic chicane, and the X-ray pulse (red) is circulating in the cavity, overlapping with electron bunches that pass the undulator at 2.23 MHz. A retractable transmission grating extracts a fraction of the circulating X-rays for diagnostics. Cerium-doped yttrium aluminium garnet (Yag:Ce) scintillating screens can be inserted at both sides of the cavity for photon alignment.

repetition rate of 2.23 MHz by translating the downstream unit. As intra-cavity diagnostics, an X-ray transmission grating was inserted into the beam path that diffracts a small fraction of the beam onto a photodiode. In addition, a transmissive scintillating screen, a single-shot spectrometer and an X-ray gas monitor located downstream of the cavity were used to monitor the outcoupled X-ray pulses. Details of the set-up and alignment procedure are described in Methods. A list of the most important parameters is given in Extended Data Table 1.

First, the cavity was spatially aligned to a condition, where a 'cold' ring-down of individual pulses in the cavity can be observed (Fig. 2, orange curve). In this case, trains of 100 250-pC electron bunches at a repetition rate of 2.23 MHz were injected into the cavity. Without matching the cavity length, no amplification occurred, and the radiation from a single pass of electrons through the undulator was bouncing inside the cavity, without interaction with further electron bunches. With femtosecond synchronization and micrometre-scale transverse overlap of the electron bunches and the recirculating, monochromatized photon pulses, they seed the successive electron bunches, thereby increasing photon production within a narrow bandwidth. To achieve this synchronization, the length of the spatially aligned cavity was scanned on the micrometre-scale while monitoring the pulse energy of the recirculating photon pulses using grating-based in-cavity diagnostics as well as the spectrometer and the downstream transmissive imager. In Fig. 3, the cavity length variation is shown on the vertical axis and the horizontal axis shows the recirculating pulses. When the cavity length matches the repetition rate of the electron bunches, a strong increase of the spectrometer signal from bunch to bunch is observed. This was accompanied by a strong rise in signal on the downstream transmissive scintillator as well as saturation of the in-cavity diagnostics (Extended Data Fig. 1). The apparent dependence on the cavity length is conclusive evidence that the oscillator lasing condition has been achieved.

Figure 2 shows the ring-down of the diode signal in seeding condition (blue curve, grating retracted) as well as for the cold-cavity case (orange curve, grating inserted). From these curves, the cumulative cavity reflectivity for a single round-trip is estimated to be 67.3(4)% for the case with seeding and 75% for the cold cavity (corrected for the grating losses). This is significantly lower than the theoretical combined peak reflectivity of 96.8%. As discussed in literature[39], such differences can be attributed to imperfections in the optical components, such as strain in the crystals that reduce the peak reflectivity. In addition, wavefront distortions introduced by the imperfections of mirrors and crystals lead to higher cut-off losses at the finite-length mirror apertures. In addition, imperfect alignment of the mechanics can lead to stronger losses[38]. However, this effect is suspected to be small, as the cavity was aligned by maximizing the ring-down signal. In the simulation (Methods), the ring-down is already reduced to 85% by including the finite size of the mirrors and their measured imperfections.

Spectra in lasing condition taken downstream of the cavity are shown in Fig. 4. The photon spectrum generated by the first bunch shows a very broad distribution with a bandwidth of $E_{FWHM} = (21 \pm 6)$ eV full-width at half-maximum (FWHM), corresponding to a unsaturated SASE spectrum from a single pass. The later bunches, exemplarily shown for the 65th as the last one measured by the spectrometer, produce much narrower and more intense spectra as a result of the seeding from the

**Fig. 2 | Photodiode ring-up and ring-down traces. a,b,** Baseline-corrected photodiode traces averaged over 1,000 pulse trains showing ring-up (first 30 pulses) (**a**) and ring-down (pulse at the last electron bunch and 20 pulses after) (**b**) of the seeded (blue) X-ray pulses. The pulses are normalized to the peak of the first bunch. The vertical dotted lines indicate the bunch number. For comparison, the signal without seeding is shown in orange. The actual photodiode signal of both traces are not comparable, as the orange curve was taken with grating inserted and the blue one with grating retracted (98% lower base signal for the blue curve; Methods). For the blue curve, the first bunches show a double peak structure, where the second peak is caused by the subsequent electron bunch passing by. The $\Delta t \approx 267$ ns separation between the minor and the next dominant peak corresponds to the round-trip light path to the downstream crystal and back to the photodiode. This is only visible owing to the lower base signal of the blue curve. For the ring-up and the pulses 101 to 103 in the ring-down, the blue curve is affected at large signal rates by detector saturation, showing as a flattening of the peaks. In the ring-down, the orange line has additional losses due to the grating.

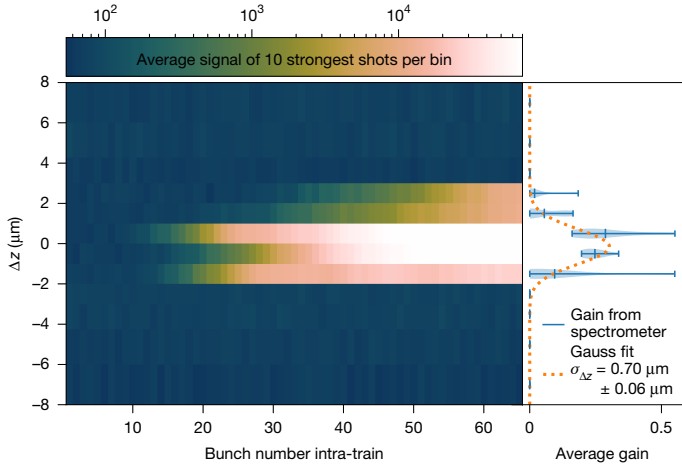

**Fig. 3 | Cavity resonance scan.** Trace of the peak spectrometer signal on a logarithmic scale during a longitudinal cavity scan with 1-μm step width. Per step, the average of the ten most intense trains is shown. The $x$ axis shows the bunch number within an electron bunch train, with the bunches having a 2.23-MHz spacing; the $y$ axis shows the cavity delay detuning $\Delta z$. Near resonance ($\Delta z \approx 0$), an orders-of-magnitude increase in the spectral signal is visible, indicating lasing. Off-resonance, a three-orders-of-magnitude-weaker and stable signal without gain is observed. On the right, the calculated round-trip gain (Methods) is shown as a violin plot over the ten samples per step. The Gauss fit shows that seeding occurs over a longitudinal range of $\sigma_z = (0.7 \pm 0.06)$ μm.

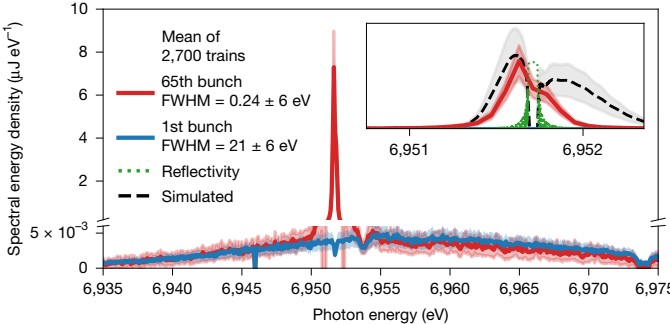

**Fig. 4 | Spectral profile narrowing.** Spectra taken downstream of the cavity set-up. The main panel shows the spectrum generated by the 1st bunch in blue and by the 65th bunch in red. The $y$-axis scale is broken to highlight both high and low spectral energy densities. The dip in the blue curve at a photon energy $E_{ph} = 6,951.7$ eV is caused by the reflection of the downstream crystal. The dip to its right is caused by another diamond-based beamline component. The inset emphasizes the spectrum of the 65th bunch with a zoomed-in view. The green dotted line in the inset shows the calculated, idealized reflection curve of the downstream crystal. The black dashed line in the inset shows a sample simulation curve based on the experimental parameters (Methods). The more pronounced right wing of the simulated curve can be traced back to a difference in electron phase space between simulation and experiment.

recirculating photon beam in the cavity. The inset in Fig. 4 shows that the seeded spectrum has a certain asymmetry. This can be understood from the very simple outcoupling mechanism used in this demonstrator set-up: instead of using a dedicated outcoupling mechanism (for example, as proposed in refs. 28,30,40), the photons transmitted through the downstream crystal were monitored[7,10]. The central part of the spectrum is reflected inside the cavity (see green dotted curve in Fig. 4). Therefore, only the wings of the spectrum generated by a single pass through the undulator are transmitted in this scheme. This leads to a double-hump structure in the spectrum of the transmitted pulse. Owing to the limited resolution of the spectrometer, which is $\sigma_E \approx 60$ meV rms (Methods), it appears here as an asymmetric curve of FWHM $E_{FWHM} = (0.24 \pm 0.06)$ eV.

By referencing the spectrometer to a calibrated gas-based X-ray pulse energy monitor[41] downstream of the set-up, the per-pulse energy evolution can be determined. The maximum pulse energies measured during a pulse train range from 4 μJ to 13 μJ, varying from train to train, with an averaged observed gain $\overline{\mathcal{G}}$ per round trip of 0.4 (Methods and Extended Data Fig. 2b). Considering that only the outcoupled fraction is measured, one can expect the number of photons stored inside the cavity to be higher, by an amount varying with the evolution of the intensity[10,39] and not quantified during this experiment. Limitations of the current set-up by heat load effects are shown in Fig. 5, where the integrated spectrum of the outcoupled photon pulse was measured with the spectrometer as a function of bunch number. In this case, the diagnostic grating was retracted to maximize gain. One can see that for three sampled individual bunch trains, the pulse energy reaches a maximum, then decays and oscillates back to some average value. Owing to the long oscillation period over multiple bunches, this cannot be explained by shot-to-shot fluctuations in the beam quality. Also, saturation of the FEL process is expected at much higher pulse energies at around 1 mJ (refs. 10,39). Compared with the simulation (see the black dashed line) and as discussed in the literature[10,25,39,42,43], this can be attributed to heat load effects on diamond crystals, which affect the FEL process in two ways. First, the non-homogeneous temperature distribution leads to heat bumps on the lattice, decreasing the overlap of the reflected photon pulses with the electron beam. Second, the

seeded and intensified incident X-rays lead to different temperature drifts for both crystals in the range of a few kelvin during a pulse train. At some point, the lattice constants are too different because of thermal expansion and the reflection curves of the two crystals do not overlap anymore. Then the photon field decays, resulting in a temporal decrease of heat load and, thus, to a response loop causing the pulse energy to oscillate. As shown in simulations[10,39,42], this effect can be strongly reduced by cooling the diamond crystals to cryogenic temperatures, bringing about a significantly higher photon output. However, with the current set-up, the anticipated increase in thermal load will probably also cause instabilities in the low-temperature regime[10].

In conclusion, we have demonstrated lasing with multi-pass gain in a CBXFEL. This achievement provides definitive experimental proof of concept for CBXFELs, two decades after their initial theoretical proposal[7,8]. Our results establish the feasibility of synchronizing a 132.8-m X-ray cavity with 2.23-MHz-repetition-rate electron bunches, overcoming challenges in alignment, stability and synchronization.

The observed X-ray pulses show high spectral purity and narrow bandwidth, confirming the key theoretical predictions[7,10], and demonstrate the potential of CBXFELs to deliver fully coherent X-ray radiation in a relatively compact assembly. Although the current proof-of-concept set-up has not yet reached the peak intensities of state-of-the-art SASE or self-seeded XFELs, significant improvements are expected through further optimization of electron beam and cavity parameters, cryogenic cooling of the crystals[10,42], and advanced outcoupling schemes[30,44]. Simulations suggest a ten- to hundred-fold increase in pulse energy up to millijoule levels and an increase of up to three orders of magnitude in spectral flux, surpassing the capabilities of self-seeding and SASE[10,28,30,39]. Further experiments are planned to investigate this.

This demonstration of X-ray lasing in a cavity opens the path towards next-generation X-ray sources with unprecedented coherence, stability and spectral brightness. The CBXFEL concept enables transformative experiments such as millielectronvolt-resolved inelastic and nuclear resonance spectroscopy[45], ultrafast pump–probe resonant inelastic X-ray scattering[46] in the hard X-ray regime, and nonlinear wave-mixing[47,48], while unlocking regimes in quantum X-ray optics[49]. These capabilities not only expand the frontiers of X-ray science but also motivate more compact and user-friendly facilities delivering high-coherence beams to a broader experimental community.

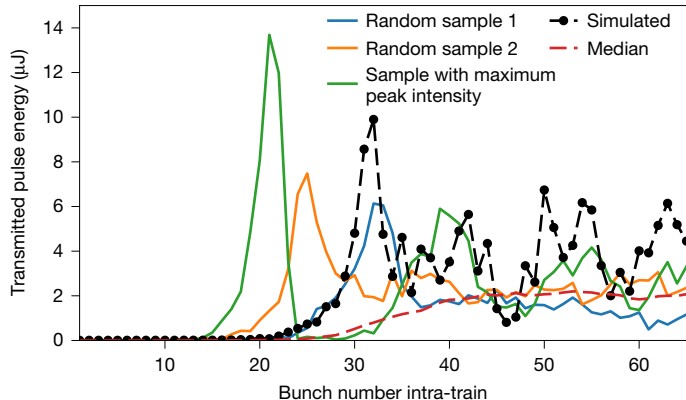

**Fig. 5 | Intra-train intensity evolution.** Evolution of integrated spectrum versus bunch number. The different colours indicate individual sample trains. The different onsets and maxima of the curves are owing to the different gain varying from train to train. The dashed red curve shows the median signal per intra-train bunch number over the 2,700 trains in the dataset. It shows much smoother behaviour as the oscillations are mostly washed out by the statistics. The dashed pointed black line is an exemplary simulation with the parameters taken from the experiment (same as in Fig. 4). Extended Data Fig. 2a shows the statistical distribution of the pulse energy taken over 2,700 trains in the dataset.

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

# Methods

## The European XFEL accelerator

The European XFEL is an XFEL for soft and hard X-rays, covering photon energies in the range 0.25 keV to 25 keV (ref. 5). The photons are generated in 3 undulator lines (SASE1, SASE2 and SASE3)[5], which serve, in total, 7 experimental stations. The European XFEL is operated in pulsed mode, with up to 2,700 electron bunches within one 600-μs-long radio-frequency-pulse being accelerated by a superconducting accelerator. The radio-frequency pulses repeat every 100 ms and provide up to 4.5-MHz electron bunch repetition rate inside a pulse[5]. These pulses are referred to as electron bunch trains in the main text. First lasing and first user operation in the SASE1 undulator line was achieved in 2017[5,50]. The 213.5-m-long SASE1 undulator line, where the experiment is located, makes use of in total 35 variable-gap undulator segments, each 5-m long. The segments are separated by 1.1-m sections filled with quadrupole magnets for focusing, electron beam position monitors, permanent magnet phase shifters and beam-correction coils[5,51]. The variable undulator gap in combination with variable electron beam energies $E_{beam}$ = 8 GeV to $E_{beam}$ = 17.5 GeV allows adjustment of the photon energy $E_{ph}$ in the range of 3 keV to 24 keV (ref. 51). The experiment ran at 14 GeV.

## X-ray cavity

As shown in Fig. 1, the cavity is composed of two 100-μm-thin diamond crystals with ⟨100⟩ surface cut produced by TISNUM, similar to the ones used at the European XFEL self-seeding set-up[52]. The diamonds are used under back diffraction for the (400) Bragg reflection and are 66.41 m apart. They are each combined with a pair of orthogonal 70-mm-long Kirkpatrick–Baez mirrors[53] with a boron carbide ($B_4C$)[54] coating and a meridional curvature with a radius of ~30 km with coating and mounting errors. The focus is numerically optimized and chosen so that after convex bending of the crystal surface by heat load, the cavity remains in a stable condition. The diamond reflection planes are positioned at an incident angle of $\Theta = \frac{\pi}{2} - 4.438$ mrad with respect to the incident radiation. The close-to-normal incident angle defines a reflection photon energy of $E_r$ = 6.9517 keV. At each end of the cavity, the Kirkpatrick–Baez mirror and diamond surfaces are aligned perpendicular to each other, creating a retro-reflector that is mostly insensitive to angular distortion and significantly helps to stabilize the alignment on a nanoradian level. After reflection, the returning beam has a distance of only 2.67 mm from the incident beam and fits into the same vacuum pipe. The electron beam is coupled in via a 5-m-long magnetic chicane, producing an offset of 4 mm at the crystal position, with a comparably small $R_{56}$ = −22 μm, which does not deteriorate the beam.

In contrast to laser-oscillators in the visible, the longitudinal mode content of the CBXFEL is defined by the bandwidth of $\sigma_E \approx 50$ meV of the diamond (400) reflection and the longitudinal phase space of the electron beam in combination with the FEL gain.

## Electron beam optimization

The 250-pC electron beam was tuned to minimize energy spread and emittance, as is done for regular FEL operation. The electron beam compression settings were chosen to balance peak current and low energy chirp to reduce redshift of the lasing[30] while providing high FEL gain. This was done using a passive corrugated wakefield streaker after the SASE2 line[55] (Extended Data Fig. 3) as well as the X-ray spectrum in SASE mode. As noted in Extended Data Table 1, the peak current was at $I_p \approx 6$ kA at a root mean square bunch length of $\sigma_s$ = 6.5 μm with a chirp of $\frac{dE}{ds} \approx (4 \pm 0.5)$ MeV μm$^{-1}$ in the central part of the bunch, where $s$ is the longitudinal bunch coordinate. The orbit of the electron beam within the four undulator segments was set to a straight trajectory as follows. A reduced number of segments is optimized in the linear gain regime of the FEL at a pulse energy

around $Q_{SASE} \approx 100$ μJ, starting from an optimized section in the middle of the 35 segments of the SASE1 undulator line. By opening an upstream and closing the following downstream segment and re-optimizing, this section is slowly shifted downstream until the electron beam trajectory within the CBXFEL undulator cells is aligned with an offset smaller than ±10 μm. This trajectory deviation is small enough for sufficient overlap with the circulating X-rays. The FEL gain measured in SASE mode with 9 additional upstream cells yielded for the last 4 cells an integrated value of $G_{FEL}$ = 48, corresponding to a gain length of $L_G \approx 5$ m. It should be noted that the total integrated FEL gain in SASE mode and in the seeded mode are very different. This is mostly owing to the so-called lethargy regime, which defines the roughly first two gain lengths in an undulator line that do not yield gain. But also the different durations and possible transverse misalignment between the seeding pulse and the shorter electron bunch reduce the integrated gain for the seeded mode.

## Cavity alignment and synchronization

Diamond crystals and the Kirkpatrick–Baez mirror pairs are each mounted on custom-designed nanometre/nanoradian-scale precision stages providing four degrees of freedom. Alignment was performed stepwise, observing the response of the individual components' motions on insertable scintillator screens monitored by two-dimensional charge-coupled device cameras at the upstream retro-reflector assembly and 5 m upstream of the downstream one. The alignment was finalized by maximizing the cold-cavity ring-down signal using a photodiode. The projected mirror apertures of (200 μm × 200 μm) imposed stringent angular and translational tolerances, requiring submicroradian and submicrometre precision.

Synchronization of the cavity round-trip time with the electron bunch repetition rate was achieved by translating the downstream crystal-mirror assembly in single-micrometre steps over a range of 6 mm. The cavity was considered in resonance when the photodiode showed a persistent, saturated signal over the electron bunch train.

## Spectral and temporal diagnostics

For measuring the pulse-resolved intra-cavity intensities, a silicon avalanche photodiode in combination with an electronic amplifier, a pulse stretcher and a fast analogue-to-digital converter with a temporal resolution of 9.231 ns was used. A transmission diamond grating with 83% transmissivity coupled out 2.8% in the first-order diffraction mode to the diode, with 73% remaining in the zeroth order. This diagnostic can pick up the individual pulses in a pulse train of the European XFEL accelerator down to very low pulse energies. The grating could be removed to minimize losses in the cavity. In seeded mode, the higher-order diffraction modes originating from the mirror aperture carried a sufficient number of photons to be measured by the diode, making it possible to measure seeding intra-cavity without inserting the grating at a fraction of $6 × 10^{-4}$ of the full power. During seeding, the signal on the photodiode was too high and saturated the analogue-to-digital converter, so only the first pulses in the ring-up and the ring-down after the last electron bunch can be quantified. For future experiments, the raw diode signal can be adjusted to avoid saturation by appropriately tailoring the front-end electronics. In the demonstration reported here, there was not sufficient time to optimize these parameters.

The spectral properties of the X-ray pulses were characterized using the High Resolution Hard X-ray single-shot (HIREX) bent crystal spectrometer[56], which provides single-shot, high-resolution spectra with up to single-photon sensitivity and dynamic gain over four orders of magnitude with nanosecond gain switching.

## Pulse energy evolution and measurement of round-trip gain

The cavity pulse energy build-up was determined from the HIREX spectra. For each electron bunch $n$ in a train, the integrated spectrum was recorded and referenced to a pulse energy:

$$Q_{tr}(n) = \int S_{tr}(E_{ph}, n) dE_{ph}$$
$$\approx \mu\alpha \sum_i \widetilde{S}_{tr}(E_{ph}^i, n)\Delta E_{ph}, \quad \text{where } E_{ph}^i = E_{ph}^{i-1} + \Delta E_{ph} \tag{1}$$

$$\text{and } \widetilde{S}_{tr}(E_{ph}^i, n) = \int S_{tr}(E_{ph}, n) e^{-\frac{(E_{ph}^i - E_{ph})^2}{2\sigma_E^2}} dE_{ph}$$

where $S_{tr}$ is the real transmitted spectrum and $\widetilde{S}_{tr}(E_{ph}^i)$ is the discrete signal measured by the HIREX spectrometer at the $i$th pixel of the strip detector. The spectrometer has a bin size of $\Delta E_{ph} = 0.08$ eV and a Gaussian resolution kernel of $\sigma_E = 61$ meV. $\alpha$ is a variable attenuation factor and $\mu$ is a calibration factor from the spectrometer to a real spectral energy density.

From the evolution of the pulse energy, a round-trip gain can be determined:

$$\mathcal{G}_n = \frac{\widetilde{Q}_{tr}(n+1)}{\widetilde{Q}_{tr}(n)} - 1. \tag{2}$$

To obtain better precision on the seeding gain, for the calculation of the round-trip gain, the integral over the pulse energy was restricted to a range $\widetilde{Q}_{tr} = \sum_{\pm 1eV} \widetilde{S}_{tr}(E_{ph,i} - E_r)$ around the resonant photon energy $E_r$. The round-trip gain $\overline{\mathcal{G}}$ was obtained as the median over the average per train in all seeded bunches in the linear gain regime. The latter was selected by defining a lower limit $\widetilde{Q}_{tr} \geq 5$ nJ to eliminate the spectral noise, and an upper limit of $\widetilde{Q}_{tr} \leq 0.5$ μJ to remove the pulses already witnessing thermal saturation effects. A histogram over the calculated gain per train is shown in Extended Data Fig. 2.

## Data analysis

Photodiode traces were averaged over many trains and baseline-corrected using asymmetric least square smoothing. To refer the spectrometer to actual pulse energies, the photo-ion signal of an X-ray gas monitor[41] was used. This signal is calibrated to absolute numbers and, after letting it stabilize for several seconds, provides the average photon flux with 9% uncertainty. A constant scaling factor for the spectrometer was then derived by comparing the average integral from the spectrometer signal to the X-ray gas monitor reading. For a complete evolution of the mean spectra over the bunch train, spectra with low and high attenuation were combined, where for spectra with values $\widetilde{S}_{tr}(E_{ph}^i) \leq 23$ nJ eV$^{-1}$ either the low attenuation spectral content was used or the high attenuation content with their (aforementioned) respective attenuation factor $\alpha$.

## Simulation

Simulations were performed to compare the experimental results with the theoretical predictions. The simulations intertwine the FEL generation code Genesis-1.3[57] with the self-developed wavefront propagation code pXCP[58] and the commercial finite-element software Comsol Multiphysics for modelling the thermal response. The framework is described in detail in ref. 39. Compared with previous simulations presented in ref. 10 for the experimental set-up, some parameters were adjusted. The mirror surfaces were modelled after the profiles measured in the laboratory and the energy chirp was matched to the one measured during the experimental campaign. The undulator magnetic-field strength was treated as an adjustable parameter, enabling a slight shift of the peak of the SASE spectral envelope away from the reflection curve by a fraction of the SASE bandwidth. This adjustment reduced the seeding gain to replicate the experimentally observed round-trip gain and to account for additional, unresolved sources of gain reduction, such as imperfections in the electron beam distribution. In addition, a $\Delta T = 2$ K temperature difference between the upstream and downstream crystal was measured and implemented into the simulations. With these adjustments, the simulations agree quite well with the experiment. The strongest deviations can be observed in the spectral shape, where the simulated spectrum shows a more symmetric shape around the reflection curve, and in the ring-down, with the simulations yielding a higher cavity reflectivity of 85%. These deviations can be easily explained by the uncertainties in the experimental parameters, such as the actual six-dimensional electron phase space, the unknown static strain in the crystals and the specific alignment accuracy of each X-ray optical component. It should also be noted that by adjusting the undulator gain, the strength of the thermal oscillations could be influenced, with a larger gain yielding a stronger oscillation amplitude. However, this can also be observed in the experiment, with the samples with high gain exhibiting the largest peak fluence (see, for example, the green curve in Fig. 5).

## Data availability

The raw and processed experimental data supporting the findings of this study are available via Zenodo at https://doi.org/10.5281/zenodo.17630323 (ref. 59). Source data are provided with this paper.

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

**Acknowledgements** We acknowledge the support from the European XFEL GmbH, Schenefeld, and DESY, Hamburg, Germany. This work was partly funded by the European XFEL research and development programme, beam time was provided within the European XFEL facility development programme, and computational resources were obtained from DESY's Maxwell cluster. We thank the involved technical groups of the European XFEL and DESY for their assistance, in particular MVS, MDI, MEA2 and U. Brüggmann; D. Lipka for the work on the diagnostic imager station inside the X-ray cavity; K. Tasca for characterization of the diamond crystals using Rocking Curve Imaging at BL05 at ESRF; and M. Störmer for making the B₄C coatings for the mirrors. B.F. and D.L.C. thank T. Noll for his input on high-precision optomechanics. P.R. thanks S. Tomin for help on analysing the electron bunch longitudinal profile; and Z. Huang and A. Halavanau for discussions on cavity-based X-ray FELs and the experimental set-up.

**Author contributions** All authors were part of the collaboration that conceived of, planned and built the set-up, and performed the experiment. The concept of a cavity-based X-ray FEL at the European XFEL was conceived by W.D., J.R. and H.S. The retro-reflector set-up was conceived of and developed by H.S., B.F., D.L.C., M.D.F., F.Y., P.R. and I.B. The adaption of the electron beamline was conceived of and implemented by W.D., P.R., T.W. and L.M. The technical infrastructure development was done by B.R., D.T., M.D., S.K., S.C., A.S. and V.S. Control system integration and online analysis tools were developed by F.M., I.K. and A.S. Metrology and assembly of X-ray optics was carried out by M.V., S.S., I.F.M., L.S. and I.B. Photon diagnostics was developed by J.G., W.F., A.K., N.K., J.L. and T.M. The experiment was carried out by P.R., I.B., B.F., W.F., J.G., M.G., A.K., N.K., S.K., D.L.C., J.L., M.M., J.R., L.S., M.S., H.S. and M.V. The data evaluation was done by P.R. The paper was written by P.R. and H.S. and reviewed by all authors.

**Funding** Open access funding provided by Deutsches Elektronen-Synchrotron (DESY).

**Competing interests** The authors declare no competing interests.

**Additional information**
**Correspondence and requests for materials** should be addressed to Patrick Rauer.

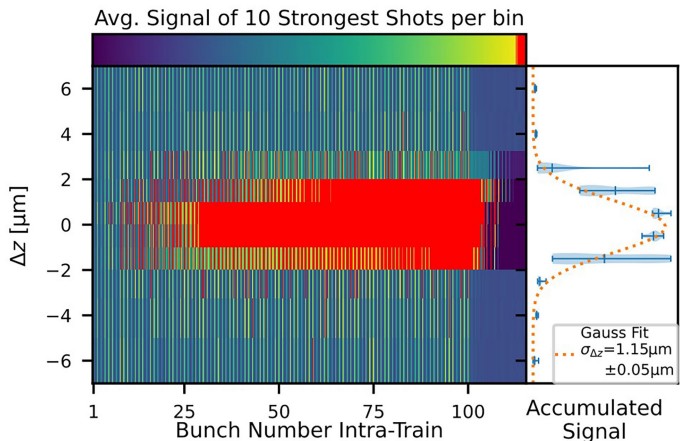

**Extended Data Fig. 1 | Resonance scan with detector saturation.** Photodiode signal trace in false color coding during a longitudinal cavity scan with 1 μm step-width. Per step, the average of the ten most intense trains is displayed. The x-axis shows the bunch number within an electron bunch train, with the bunches having a 2.23 MHz spacing; the y-axis the cavity delay detuning Δz. Near resonance (Δz ≈ 0), the diode signal is strong, sustains and exhibits clear saturation (red color), indicating lasing. Off-resonance, only weak, rapidly decaying signals are observed. After the 100st bunch, a faint ring-down can be observed in the signal. For the cases when the diode is saturated, the signal rapidly switched to a constant -32,000 after the impinging power fell below a threshold. On the right, the sum over the diode trace (see Methods) is shown as violin plot for the 10 samples per step. The gauss fit shows seeding over a longitudinal range of $\sigma_z = (1.15 \pm 0.05)$ μm This is slightly larger than when mapping against the round trip gain. This is due to the early saturation of the diode smearing out the actual response against detune.

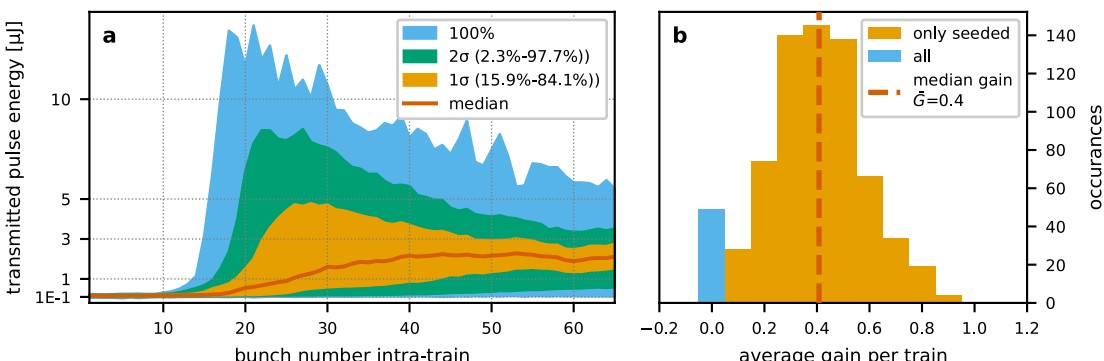

**Extended Data Fig. 2 | Spectral evolution statistics. a** shows the median and three different quintiles (1σ, 2σ and 100%) over 2700 trains of electron bunches, with the statistics evaluated on a intra-train bunch basis. **b** shows the histogram of the average gain per train, evaluated as described in the Methods. The orange block only includes seeded shots, and was used for the median calculation. The blue blocks also includes the non-seeded shots, which are 7% of all trains.

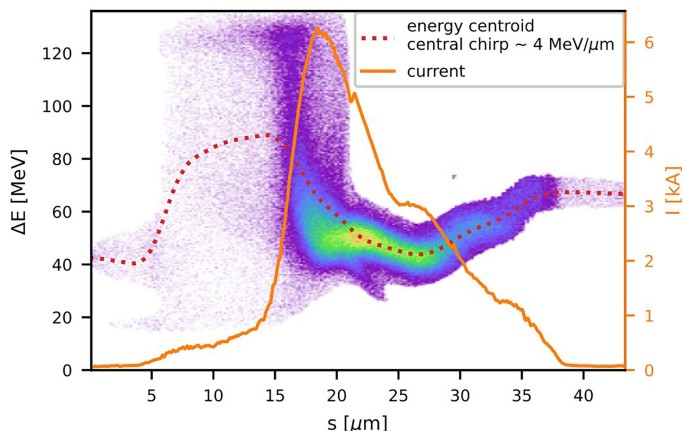

**Extended Data Fig. 3 | Electron phase space.** Reconstructed longitudinal phase space of the electron beam used in the experiment. The measurement was done after the SASE2 line using a passive wakefield streaker[55].

**Extended Data Table 1 | Relevant electron beam and X-ray cavity parameters**

| Electron beam and FEL parameters | | | | | |
|---|---|---|---|---|---|
| Beam energy | Bunch charge | Peak current | Bunch length | Central chirp | energy jitter |
| 14 GeV | 250 pC | 6.2 kA | 6.5 µm rms | $\sim$4 MeV µm$^{-1}$ | 1.4 MeV rms |
| arrival time jitter | position jitter | pointing jitter | Undulator length | Gain length | |
| 10 fs rms | 3 µm rms | 100 nrad rms | 20 m | $\sim$5 m | |

| Cavity parameters | | | |
|---|---|---|---|
| Photon energy | Crystal cut | Reflection order | Crystal incidence angle |
| 6.952 keV | <1 0 0> | (4 0 0) | 4.38 mrad |
| Mirror length | Mirror curvature | Mirror surface roughness | Mirror grazing angle |
| 70 mm | $\sim$30 km | <1 nm | 3.1 mrad |
| Cavity length | e-beam offset by chicane | distance X-ray beams before and after reflection | |
| 2×66.41 m | 4 mm | −2.67 mm | |