## [Peer Review file · Nature]

Lasing of a Cavity Based X-ray Source

Corresponding Author: Dr Patrick Rauer

Version 0:

Reviewer comments:

Referee #1

(Remarks to the Author)

The main argument of this paper is that it demonstrates, for the first time, multi-pass gain lasing in a cavity-based X-ray free electron laser (CBXFEL). This finding confirms the high spectral purity that was predicted by theory 20 years ago. It shows that a CBXFEL can produce fully coherent X-ray radiation using a relatively simple configuration.

The LCLS-II CBXFEL, which was recently published in Nature Photonics, employs a grating for diagnosing a ring-down XFEL. Unlike the LCLS-II CBXFEL that utilizes four Bragg mirrors as the primary cavity and a single beryllium lens at the center of the cavity, this experiment significantly differs by employing four undulators as the amplification medium between two retro-reflector cavities made of crystals and KB mirrors. The grating-based diagnostic method is used in this setup.

The paper experimentally confirmed lasing and demonstrated spectral purity through seeding. Although the transmitted pulse energy remains at the level of several tens of microjoules, indicating room for improvement, this research is valuable as it illustrates the feasibility of CBXFEL and includes an analysis of the crystal heat load effect.

This paper reports that the measured pulse energy in the pulse train ranged from 4 to 13 μJ . It notes that only a portion of the stored photon energy was outcoupled, and the oscillations observed in the time evolution of the transmitted pulse energy can be explained by thermal effects predicted by theory.

The authors also suggest that cooling the crystal to cryogenic temperatures could increase the output by 10 to 100 times, potentially surpassing both self-seeding and SASE (Self-Amplified Spontaneous Emission). However, the reference [31], which discusses the simulation review of this demonstration, indicates that the beam quality deteriorates as the pulse energy increases. Therefore, a more detailed explanation of this prediction is needed.

In line 98, the term “micrometer accuracy” could be misleading. A cavity length offset of 1 micrometer will result in a 100 micrometer offset after 100 round trips, which is considerably larger than the bunch lengths in HX-FEL, which range from 10 to 20 micrometers. The inaccuracy in cavity length is directly related to the gain instability discussed in this paper.

Additionally, the cavity length detuning results presented in Fig. 3 (which is also the primary focus of Fig. 2) are based on measurements from a saturated detector, making it challenging to compare the detuning caused by variations in the gain curve. The 1000 averaged data points shown in Fig. 2 suggest a significant instability in the gain curve on a pulse-by-pulse basis. However, the statistics supporting this observation are not provided.

The beam characterization is conducted solely using leakage X-rays from behind the Bragg reflector. Unfortunately, it cannot be performed at the intended output coupler or grating due to significant detector saturation. This saturation creates ambiguity in interpreting fundamental aspects of the photon beam, including its spectrum, energy, mode (coherence), and pulse duration. Therefore, this reviewer recommends incorporating additional experimental results to enhance the analysis.

The results of the beam dynamics simulation, including the parameters for optical cavity design, will assist in understanding the findings. The gain spectrum will be redshifted as the photon energy increases. Adjusting the e-beam energy within the bunch train may help align the photon energy with the Bragg reflections.

The measured “cumulative cavity reflectivity” is lower than the theoretically predicted value of 96.8%, reaching only 67.3% in

the absence of grating. The authors attribute this lower reflectivity to two main factors: degradation of peak reflectivity due to strain within the crystal and wavefront distortion. Additionally, this observed value appears to be lower than the results from the previous LCLS CBXFEL study. In that study, misalignment was identified as a significant factor. I'm curious to hear your thoughts on the impact of this misalignment.

In conclusion, this result is significant for publication as it demonstrates the feasibility of CBXFEL using an intra-cavity undulator, in contrast to previous CBXFEL experiments.

One typo: Page 8, 357th line "retro-reflector" -> "retroreflector"

Referee #2

(Remarks to the Author)

This paper reports on the first lasing of the cavity-based X-ray free electron laser (CBXFEL), or the so-called XFEL oscillator (XFELO). In contrast to the conventional XFELs based on the self-amplified spontaneous emission (SASE), CBXFEL potentially provides fully (spatially and temporally) coherent and intense x-ray radiation, and thus may open new applications of XFELs. Although the laser power currently achieved is not outstanding compared to those achieved with conventional methods, the authors have experimentally demonstrated the concept to realize XFELO proposed more than 15 years ago. This is definitely an important milestone toward new sciences with XFELs; congratulations to the authors! I strongly recommend the paper for publication, however, the following points should be considered.

[1] Comparison with theory

Although the experimental results obviously demonstrate the lasing of CBXFEL, no discussions are found about the comparison with theoretical predictions. Comprehensive numerical simulations should be carried out to be compared with the experimental results. If the two results have a large discrepancy, possible reasons should be addressed.

[2] Quality of figures

Figure 1, which shows the schematic of the experimental layout, is not convincing. Besides the 3D drawing, 2D figures (top and side views) should be added with all the components (intra-cavity and downstream diagnostics). Personally, the 3D version is not needed.

Figure 3 is not convincing as well. What does "10 samples are equally distributed over the bin in Delta z" mean? Because this figure is the most important in this paper as evidence of lasing, more detailed explanations and better visualizations are recommended.

[3] Comparison between no-seed/seeded results

Figure 2 compares the growth and decay of the x-ray pulses with and without seeding. As far as I understand, the intensity (PD signal) is normalized so that the saturation level is similar, and thus the initial level in the ring-up pulses have 2 orders of magnitude difference. This makes it difficult to directly compare the actual intensity between the two conditions. I recommend normalization to balance the initial level.

[4] Interpretation of the experimental results

Two points should be discussed about the experimental results. First, Fig.3 shows that there exist a few conditions (Delta z, or the cavity length) that significantly enhance the lasing; e.g., Delta z = -0.5um, 0, 1um. Second, Fig. 2 shows that the initial ring-up pulses (1-10) have a characteristic double-peak structure, which is not found in later pulses. The authors should explain the reasons for the above phenomena.

[5] Experimental condition

I have an impression that the experimental condition is not clearly shown or the information is insufficient. For improvement, a table to summarize important parameters regarding the electron beam, undulator, and crystal should be inserted.

Other information to be added:

- What is the offset of the electron beam to insert the retro-reflector? Does the resultant R56 have any impact on the electron beam?
- What is the fraction (percentage) of the beam to be diffracted and transferred to the photodiode for intra-cavity diagnostics?

[6] Bandwidths of FELs

In line 076, the authors mention that the typical bandwidth of SASE radiation is about 1%, which I agree. In general, however, the bandwidth is theoretically given by rho (Pierce parameter) as presented in [9] (and other texts), and is of the order of 0.1% in most XFEL facilities. The discrepancy may come from the energy chirp (and jitter) of the electron beam. The authors should address this issue to avoid misunderstanding.

[7] Minor revisions and typos

- line 187, "E_{fwhm}=21(6) eV full width half maximum(fwhm)": what does the number (6) in parentheses mean?
- Caption of Fig. 4: "by the first bunch in blue the 65 th bunch in red"
- line 398 (exponent should be negative)

Referee #3

(Remarks to the Author)

Dear Editor,

In their manuscript, Dr. Rauer and colleagues report a recent experimental demonstration of x-ray amplification obtained with a cavity based FEL. This is an important milestone in the development of future coherent x-ray sources, and the work certainly merits being published in a high-impact journal with broad readership. I therefore recommend publication of the manuscript in Nature after a revision based on the following comments.

The manuscript is in general well written and with proper citation of previous works with the exception of earlier studies on oscillator FEL in the visible and IR. While the wavelength ranges and associated technologies differ there are also relevant similarities and common challenges (e.g., pulsed gain due to short bunches, the response of cavity optics to high radiation power, ...). These lessons from the optical and IR domains should not be overlooked by the x-ray FEL community.

It appears that in some cases different names are used for the same (or very similar) concept. This is the case for the used setup referred as a Cavity Based X-ray FEL but elsewhere in the manuscript and in the cited literature is also called XFEL (X-ray FEL Oscillator). It would be useful if in the introduction authors clarify this and then stick to one option for the whole manuscript (or explain the difference between the two).

Line 90: When referring to the need of high repetition rate beams for supporting lasing in short cavities authors should emphasize that for X-ray this needs to be combined with high electron beam energy and brightness not compatible with methods used in the past for visible or IR (synchrotron, energy recovery linac, ...) oscillator FELs.

When introducing the experimental setup in the Methods section, Authors only generally introduce the accelerator without giving any details of the parameters used for this specific case. Even the electron beam energy is not specified. Details of the electron beam compression are also very vaguely described in line 342 without any indication of the estimated peak current and bunch length. If peak current may be not critical and would only affect the gain, an estimation of the bunch length is important together with the estimated bunch arrival jitter since it may play a role in the observed intra-train oscillation of the pulse energy (Fig5). Authors are encouraged to provide a table with details of beam properties and jitter (both intra train and train to train).

Figure 3 is not clear: both the figure and the caption require clarification.

The left panel reports the diode intensity in false color coding (red high intensity) as a function of the intra-bunch bunch number and the cavity detuning (Dz). According to the text all trains are supposed to have all 100 bunches, however in the image few traces show a bunched structure also above bunch number 100.

Moreover, in some cases the diode saturation appears to persist up to bunch number 110 or more. Authors should clarify and explain these unexpected behaviors.

According to the caption the cavity length has been changed with 1 μm steps.

For each value of the cavity length 10 traces corresponding to different bunch trains are shown. For sake of simplicity traces corresponding to the same Dz are stacked and appears in the plot associated to Dz values different than the used one.

Traces for Dz=0 are plotted with DZ ranging from -0.5 μm to 0.5 μm etc. If this is the case it may be recommended to put grids in the plot to isolate data with same Dz. As it is now, the plot suggests that Dz is varied continuously.

Around Dz = -4 μm it appears that only 2 traces are plotted and not 10 as mentioned in the caption.

The right panel is not clear. Caption suggests that each point should be the integral of one of the traces plotted on the left panel. If this is the case one expects to see 10 points for each Dz value (i.e. -4,-3,-2,-1,0,1...). The plot instead reports many more points suggesting again that Dz is varying continuously.

Measurements of the cavity response indicate a significantly lower cold cavity round trip reflectivity than the theoretical one. Authors may comment on the required alignments efforts for reaching this value and if an improvement can be expected with some extra effort.

In Figure 5 Authors show the measured FEL amplification over the bunch train for 3 different trains. Data show a clear amplification up to a maximum followed by a decay and few oscillations. The effect is associated to heat load effects on the cavity elements. Authors should comment if the electron beam jitter along the train (bunch timing, energy, centroid) may be also contributing to these oscillations.

In line 337 the comment of the difference between optical resonators and CBFEL is not clear. Are Authors referring to FEL oscillator operating in the visible regime or to "standard" solid state or gas lasers?

Line 340 and following. More details on e-beam properties would be beneficial. Also, an indication of the measured gain-length in SASE mode for the used e-beam could be given as a reference of the beam quality.

Line 395: Clarify whether ΔE_{ph} is the same as ΔE

Line 401: A bracket is not closed

Line 413: Does this imply that only one spectrometer pixel is used for the analysis? Please specify

Line 692: ERSF is probably ESRF

Line 706: While in the mail text the term CBXFEL has been used here XFEL is used.

Version 1:

Reviewer comments:

Referee #1

(Remarks to the Author)

Numerical simulations were conducted to compare with the experimental results, as suggested by the referees. The simulation findings, illustrated in the figures, show a strong correlation with the experimental data. The figures have been revised for clearer presentation of the experimental results.

All questions have been thoroughly addressed, and the relevant issues have been incorporated into the main text.

I strongly recommend the paper for publication.

Referee #2

(Remarks to the Author)

The authors have responded to all the comments. Now I recommend the paper for publication.

Referee #3

(Remarks to the Author)

Dear Editor,

Authors have improved their manuscript based on the received comments. All major points have been properly answered with additional information or revised figures.

The manuscript is now suitable for being published on Nature.

Few minor points remain that Authors may consider for a revision before publication.

Minor comments:

In the introduction “Applying these concepts, several large-scale X-ray FEL (XFEL) user facilities were built [5–10], ...”

If authors want to specifically refer to hard x-ray facilities, reference [5] should be removed. If authors want to refer to only SASE facilities also SXFEL in Shanghai needs to be mentioned (<https://www.mdpi.com/2076-3417/12/1/176>, <https://issf01.cas.cn/en/facilities-view.jsp?id=ff80808151cd1d840151e718b8a31488>).

Caption of fig. 2

Caption reports that 25 pulses are plotted why the figure shows 30.

The comment on the double peak feature of some pulses is not clear. Why the subsequent electron should produce a signal at 10s or 100s of ns?

Extended table

It should specify if rms is used for jitter values.

Central chirp is missing μm^{-1}

Not clear what the 4 mm refer to.

Simulations:

Authors report that “the undulator magnetic field strength was adjusted so that the round-trip gain matches the one measured in the experiments”

Is this meaning that in the simulation the undulator resonance condition has been changed to allow reproducing the experimental results? What was the offset required? From the spectra in Fig.4 one would say that the undulator resonance is ok within 5×10^{-4} .

Authors report “by adjusting the Undulator gain, the strength of the thermal oscillations could be influenced ...” If this is the case, the same should be for electron beam energy jitter, but being of 1×10^{-4} maybe this is a smaller effect.

Reponse to Referees:

We would like to thank all three referees for their thorough review of the manuscript and their many valuable suggestions. In response, we have made major revisions to Figures 1 and 2, updated Figure 3 using a newly analysed non-saturated dataset with improved binning, and carried out new simulations with parameters matching the experimental conditions (shown as black dashed lines in Figures 4 and 5).

Referee 1:

General:

„The main argument of this paper is that it demonstrates, for the first time, multi-pass gain lasing in a cavity-based X-ray free electron laser (CBXFEL). This finding confirms the high spectral purity that was predicted by theory 20 years ago. It shows that a CBXFEL can produce fully coherent X-ray radiation using a relatively simple configuration.

The LCLS-II CBXFEL, which was recently published in Nature Photonics, employs a grating for diagnosing a ring-down XFEL. Unlike the LCLS-II CBXFEL that utilizes four Bragg mirrors as the primary cavity and a single beryllium lens at the center of the cavity, this experiment significantly differs by employing four undulators as the amplification medium between two retro-reflector cavities made of crystals and KB mirrors. The grating-based diagnostic method is used in this setup.

The paper experimentally confirmed lasing and demonstrated spectral purity through seeding. Although the transmitted pulse energy remains at the level of several tens of microjoules, indicating room for improvement, this research is valuable as it illustrates the feasibility of CBXFEL and includes an analysis of the crystal heat load effect.

This paper reports that the measured pulse energy in the pulse train ranged from 4 to 13 μJ . It notes that only a portion of the stored photon energy was outcoupled, and the oscillations observed in the time evolution of the transmitted pulse energy can be explained by thermal effects predicted by theory.

[...]

In conclusion, this result is significant for publication as it demonstrates the feasibility of CBXFEL using an intra-cavity undulator, in contrast to previous CBXFEL experiments.

”

Points raised:

1. The authors also suggest that cooling the crystal to cryogenic temperatures could increase the output by 10 to 100 times, potentially surpassing both self-seeding and SASE (Self-Amplified Spontaneous Emission). However, the reference [31], which discusses the simulation review of this demonstration, indicates that the beam quality deteriorates as the pulse energy increases. Therefore, a more detailed explanation of this prediction is needed.
 - We thank the referee for pointing this out. In order to better emphasise this, following section has been added to line 269: **“However, with the current setup the anticipated increase in thermal load will likely also cause instabilities in the low temperature regime [33].”**
2. In line 98, the term “micrometer accuracy” could be misleading. A cavity length offset of 1 micrometer will result in a 100 micrometer offset after 100 round trips, which is considerably larger than the bunch lengths in HX-FEL, which range from 10 to 20 micrometers. The inaccuracy in cavity length is directly related to the gain instability discussed in this paper.
 - Such a high sensitivity to the cavity length would be expected for about high Q laser oscillators analogous to the lasers in the visible regime. However, for the current case, the tolerance for longitudinal misalignment is much more relaxed for two reasons. First, due to the Bragg-reflection at the crystal the X-ray pulse becomes considerably stretched to a length of roughly 30-40 μm . Second, the gain in the system is considerably higher than 1, meaning that we predominantly see the X-ray generated over the last two to three round trips. Also

in saturation, the comparably high losses of ~30% severely damp down the X-rays coming from the earlier round-trips. This picture is also backed by our simulations. For operation with high gain, we actually expect the tolerance towards detuning to be half (as the detuning is entering twice in the cavity length) the size of the seeding pulse. At low gain, we are much more sensitive towards the peak of the seeding pulse, reducing this tolerance to the value we observed.

3. Additionally, the cavity length detuning results presented in Fig. 3 (which is also the primary focus of Fig. 2) are based on measurements from a saturated detector, making it challenging to compare the detuning caused by variations in the gain curve. The 1000 averaged data points shown in Fig. 2 suggest a significant instability in the gain curve on a pulse-by-pulse basis. However, the statistics supporting this observation are not provided.
 - The Fig.3 was updated with the data from the downstream spectrometer, which was saturating (low damping) at much higher intensities (the old figure was shifted into the Extended data). This makes it possible to compute a round-trip gain against detuning, which suggests a tolerance of $\sigma_s=0.7 \mu\text{m}$. Unfortunately, we do not fully understand the comment regarding the apparent instability in Fig.2.
4. The beam characterisation is conducted solely using leakage X-rays from behind the Bragg reflector. Unfortunately, it cannot be performed at the intended output coupler or grating due to significant detector saturation. This saturation creates ambiguity in interpreting fundamental aspects of the photon beam, including its spectrum, energy, mode (coherence), and pulse duration. Therefore, this reviewer recommends incorporating additional experimental results to enhance the analysis.
 - We agree with the referee that more data on the X-rays in the cavity would be very insightful. Unfortunately, with the current setup as is, even with a non-saturated photodiode, solely the pulse energy evolution of the in-cavity X-rays could be mapped (from theory, out of the start-up and (thermal) saturation regime, the round trip gain in-cavity and the transmitted are the same). While further upgrades to the experimental setup are planned, this will take a considerable amount of time considering the complexity of installations in the tunnel. Further, the European XFEL is currently in a long shutdown, shifting the next possible repetition of this experiment to earliest mid of next year. Also, we want to emphasise that the goal of the experiment was the demonstration of the working concept of CBXFELs, and not to have a full characterisation. As noted above, this would have strongly complicated and delayed the experiment.
5. The results of the beam dynamics simulation, including the parameters for optical cavity design, will assist in understanding the findings. The gain spectrum will be redshifted as the photon energy increases. Adjusting the e-beam energy within the bunch train may help align the photon energy with the Bragg reflections.
 - In addition to the simulations on the setup published in ref. [33], start-to-end simulations with updated parameters have been carried out and included in the manuscript. Specifically, curves based on simulation shown in Fig. 4 and Fig.5, and described in the methods section under „Simulations“. They show comparably good agreement with the experimental results. Due to a lack of high resolution electron beam phase space diagnostics at the SASE lines at the European XFEL, we have to note that the electron phase space used in simulation is expected to deviate from the experimental one. For the second point, we agree with the referee concerning the assumption of the red shift due to crystal heating. So, adjusting the e-beam energy could in principal help. However, there are two arguments against that approach. First, the gain curve of the seeded FELs is much wider than the red shift during the train. So, adjusting the electron energy will have very little impact on the actual FEL gain and, hence, saturation intensity. The second argument is following the cause for the thermal instabilities. The gain does not break down due to the redshift, but due to different heating in downstream and upstream crystals (the downstream crystal sees the full radiation, also transmitting through, while the upstream crystal only sees the filtered radiation which has a much lower total absorption due to the decreased penetration depth). At some point, this increases the reflection losses above the gain (also see reference
6. The measured “cumulative cavity reflectivity” is lower than the theoretically predicted value of 96.8%, reaching only 67.3% in the absence of grating. The authors attribute this lower reflectiv-

ity to two main factors: degradation of peak reflectivity due to strain within the crystal and wavefront distortion. Additionally, this observed value appears to be lower than the results from the previous LCLS CBXFEL study. In that study, misalignment was identified as a significant factor. I'm curious to hear your thoughts on the impact of this misalignment.

- We agree with the referee that misalignment can have a significant impact on the cavity reflectivity. This is especially true as our system introduces two small size (200µm) apertures due to the finite size of the KB mirrors. However, the entire alignment procedure relies on the minimisation of these losses (on a step by step basis) up to a precision where we do not see any differences any more (~100 nrad).. While it is highly probable that the final misalignment will have an impact on the seeding gain due to bad overlap, it should not be apparent in the ringdown. Also, the system was frequently realigned to calibrated positions to hinder the alignment changing during the beam time.

Further, simulations (without misalignment) already suggest a reduction in reflectivity to 85% accounting for the measured imperfections in mirror profile and the aperture losses. In order to point out this meaningful question to the general reader, following section has been added to the manuscript in line 176: „In addition, imperfect alignment of the mechanics can lead to stronger losses [40]. However, this effect is suspected to be small, as the cavity was aligned by maximizing the ringdown signal. In simulation (see Methods), the ringdown is already reduced to 85 % by including the finite size of the mirrors and their measured imperfections.“

7. One typo: Page 8, 357th line "retror-eflector" -> "retroreflector"

- Fixed

Referee 2:

General:

„This paper reports on the first lasing of the cavity-based X-ray free electron laser (CBXFEL), or the so-called XFEL oscillator (XFELO). In contrast to the conventional XFELs based on the self-amplified spontaneous emission (SASE), CBXFEL potentially provides fully (spatially and temporally) coherent and intense x-ray radiation, and thus may open new applications of XFELs. Although the laser power currently achieved is not outstanding compared to those achieved with conventional methods, the authors have experimentally demonstrated the concept to realize XFELO proposed more than 15 years ago. This is definitely an important milestone toward new sciences with XFELs; congratulations to the authors! I strongly recommend the paper for publication, however, the following points should be considered.

“

Points raised:

1. **Comparison with theory**

Although the experimental results obviously demonstrate the lasing of CBXFEL, no discussions are found about the comparison with theoretical predictions. Comprehensive numerical simulations should be carried out to be compared with the experimental results. If the two results have a large discrepancy, possible reasons should be addressed.

- We agree with the referee that comparison to theory is important to contextualise the experimental findings. As suggested, start to end simulations were carried out and showed good agreement with the experimental results. They are compared to the experimental findings in updated Figures 4 (inset) and 5, as well as in line 178: "In simulation (see Methods), the ringdown is already reduced to 85 % by including the finite size of the mirrors and their measured imperfections." Also, a paragraph "Simulation" has been added to the method section in line 461, giving a more detailed view on the simulation and highlighting discrepancies.

2. **Quality of figures**

Figure 1, which shows the schematic of the experimental layout, is not convincing. Besides the 3D drawing, 2D figures (top and side views) should be added with all the components (intra-cavity and downstream diagnostics). Personally, the 3D version is not needed.

- Figure 1 has been replaced by a top view of experimental setup, also including the in-cavity diagnostic components. The out-of cavity diagnostics have not been directly included too not blow up the scale of the image, but indirectly by adding a label "to Imagers/Spectro-

meter/Gas Monitor".

The side-view has not been added. As the setup is dominantly arranged in the horizontal plane, in our opinion, a side view does not give additional information to the reader.

2.1 Figure 3 is not convincing as well. What does "10 samples are equally distributed over the bin in Delta z" mean? Because this figure is the most important in this paper as evidence of lasing, more detailed explanations and better visualisations are recommended.

- We understand the criticism on Fig. 3. It has been replaced by a map showing the downstream single shot spectrometer signal against bunch number and longitudinal detune. This has the advantage that the spectrometer (which had low attenuation for this dataset) is saturating at much higher intensities, and a round trip gain against Δz can be computed (on the right of the updated Fig.3). As for the former figure 3, per 1 μm bin the ten strongest samples are taken. However, unlike before, they have been averaged into a single line to yield a more comprehensive picture. The same was done for the old figure 3 with the diode signal, which has been shifted to Extended Data Figure 1.

3. Comparison between no-seed/seeded results

Figure 2 compares the growth and decay of the x-ray pulses with and without seeding. As far as I understand, the intensity (PD signal) is normalized so that the saturation level is similar, and thus the initial level in the ring-up pulses have 2 orders of magnitude difference. This makes it difficult to directly compare the actual intensity between the two conditions. I recommend normalization to balance the initial level.

- The original Fig. 2 shows the averaged diode signal (with baseline subtracted) without any further normalisation. The difference in magnitude in the first bunches comes from the fact that the orange line ("cold cavity" ringdown) was taken with grating inserted, and the blue line (seeding) with grating retracted. This causes a two order of magnitude difference in signal strength for the same circulating intensity.

We follow the advice of the referee to normalise to the first bunch to better highlight the ring-up. In an updated Fig. 2. To better emphasise the difference in base signal, following line has been added to the caption of Fig. 2 in line 153: "The actual photodiode signal of both traces are not comparable, as the orange curve was taken with grating inserted and the blue one with grating retracted (98 % lower base signal for the blue curve, see Methods)."

4. Interpretation of the experimental results

Two points should be discussed about the experimental results. First, Fig.3 shows that there exist a few conditions (Delta z, or the cavity length) that significantly enhance the lasing; e.g., Delta z = -0.5 μm , 0, 1 μm .

- The observations in Fig.3 can be traced back to the filtering applied on the data in the original Fig. 3, which also included all the data while the stage was moving (which disturbed alignment). After more stringent filtering, these conditional features are gone.

Second, Fig. 2 shows that the initial ring-up pulses (1-10) have a characteristic double-peak structure, which is not found in later pulses. The authors should explain the reasons for the above phenomena.

- We thank the referee for pointing out this observation. These double peaks (which are vanished for the ring down) are caused by the subsequent electron bunch passing by the photodiode. It is only visible due to the strong averaging and the much reduced base-signal of the reflected X-rays in the measurement mode with grating retracted.

To point this out the readers, in line 155 we added the following section: "For the blue curve, the first bunches show a double peak structure, where the second peak is caused by an electronic response of the subsequent electron bunch passing by. This is only visible due to the lower base signal of the blue curve"

5. Experimental condition

I have an impression that the experimental condition is not clearly shown or the information is insufficient. For improvement, a table to summarize important parameters regarding the electron beam, undulator, and crystal should be inserted.

Other information to be added:

- What is the offset of the electron beam to insert the retro-reflector? Does the resultant R56 have any impact on the electron beam?

- What is the fraction (percentage) of the beam to be diffracted and transferred to the photodiode for intra-cavity diagnostics?

- A table summarising the experimental parameters has been added to the Extended Data (Extended Data Table 1). This table is highlighted in line 123 "A list of the most important parameters is given in Extended Data Table 1." Also, the more important parameters are now mentioned at the corresponding sections in the text. Regarding the chicane, a section has been added to line 352: "The electron beam is coupled in via a 5 m long magnetic chicane, producing an offset of 4 mm at the crystal position, with a comparably small $R56 = -22 \mu\text{m}$, which does not deteriorate the beam." For the grating, a section for the efficiency can be found in line 393 in the methods: " A transmission diamond grating with 83 % transmissivity coupled out 2.8 % in the first order diffraction mode to the diode, with 73 % remaining in the zeroth order. This diagnostic can pick up the individual pulses in a pulse train of the European XFEL accelerator down to very low pulse energies. The grating could be removed to minimize losses in the cavity. In seeded mode, the higher-order diffraction modes originating from the mirror aperture carried a sufficient number of photons to be measured by the diode, making it possible to measure seeding intra-cavity without inserting the grating at a fraction of 6×10^{-4} of the full power."

6. Bandwidths of FELs

In line 076, the authors mention that the typical bandwidth of SASE radiation is about 1%, which I agree. In general, however, the bandwidth is theoretically given by rho (Pierce parameter) as presented in [9] (and other texts), and is of the order of 0.1% in most XFEL facilities. The discrepancy may come from the energy chirp (and jitter) of the electron beam. The authors should address this issue to avoid misunderstanding.

- We thank the reviewer for pointing this out. Indeed, the theoretical bandwidth is given by the Pierce parameter ρ , which usually is on the order of 0.1%. In our initial submission we had written 1‰ (0.1%), but we realize that this notation could easily be misread as 1%. To avoid any confusion, we have revised the manuscript to explicitly state 0.1%."

7. Minor revisions and typos

- line 187, "E_{fwhm}=21(6) eV full width half maximum(fwhm)": what does the number (6) in parentheses mean?

- The parentheses refers to the standard deviation. It has been changed to +- to make that clearer (also in the following).

- Caption of Fig. 4: "by the first bunch in blue the 65 th bunch in red"

- Fixed to „by the first bunch in blue **and by** the 65 th bunch in red“

- line 398 (exponent should be negative)

- Thank you. The exponent has been fixed.

Referee 3:

General:

„In their manuscript, Dr. Rauer and colleagues report a recent experimental demonstration of x-ray amplification obtained with a cavity based FEL. This is an important milestone in the development of future coherent x-ray sources, and the work certainly merits being published in a high-impact journal with broad readership. I therefore recommend publication of the manuscript in Nature after a revision based on the following comments. "

Points raised:

1. The manuscript is in general well written and with proper citation of previous works with the exception of earlier studies on oscillator FEL in the visible and IR. While the wavelength ranges and associated technologies differ there are also relevant similarities and common challenges (e.g., pulsed gain due to short bunches, the response of cavity optics to high radiation power, ...). These lessons from the optical and IR domains should not be overlooked by the X-ray FEL community.

- Yes, we agree. We added to the manuscript the following section with two new references in line 72: "In the infrared to ultraviolet wavelength regime, where high-reflectivity mirror technology was already established, FEL oscillators were developed soon after [2], and numerous such systems are now in operation worldwide [3]."
1. It appears that in some cases different names are used for the same (or very similar) concept. This is the case for the used setup referred as a Cavity Based X-ray FEL but elsewhere in the manuscript and in the cited literature is also called XFELO (X-ray FEL Oscillator). It would be useful if in the introduction authors clarify this and then stick to one option for the whole manuscript (or explain the difference between the two).
 - We changed the naming in our manuscript now consistently to 'CBXFEL' and added this clarification in line 93 of the manuscript: "Cavity-based XFELs (CBXFELs) [21-23], encompassing both high-gain regenerative X-ray Amplifier FELs (XRAFELs) [21] and low-gain XFEL oscillators (XFELOs) [22], offer an alternative solution ..."
 2. Line 90: When referring to the need of high repetition rate beams for supporting lasing in short cavities authors should emphasize that for X-ray this needs to be combined with high electron beam energy and brightness not compatible with methods used in the past for visible or IR synchrotron, energy recovery linac, ...) oscillator FELs.
 - We modified the text in line 98 correspondingly: "Realizing such a device requires accelerators with sufficiently high peak brightness, capable of delivering at least tens of electron bunches at a repetition rate high enough to confine the cavity length to a practical scale of several tens of meters".
 3. When introducing the experimental setup in the Methods section, Authors only generally introduce the accelerator without giving any details of the parameters used for this specific case. Even the electron beam energy is not specified. Details of the electron beam compression are also very vaguely described in line 342 without any indication of the estimated peak current and bunch length. If peak current may be not critical and would only affect the gain, an estimation of the bunch length is important together with the estimated bunch arrival jitter since it may play a role in the observed intra-train oscillation of the pulse energy (Fig5). Authors are encouraged to provide a table with details of beam properties and jitter (both intra train and train to train).
 - We followed the referees suggestions, and added a table with important experimental parameters in Extended Data Table 1. Also, more details on the beam parameters are now given in the manuscript, and especially, in the paragraph on Electron Beam Optimization. Also, the measured longitudinal phase space is shown in Extended Data Figure 3.
 4. Figure 3 is not clear: both the figure and the caption require clarification.

The left panel reports the diode intensity in false color coding (red high intensity) as a function of the intra-bunch bunch number and the cavity detuning (Dz). According to the text all trains are supposed to have all 100 bunches, however in the image few traces show a bunched structure also above bunch number 100.

Moreover, in some cases the diode saturation appears to persist up to bunch number 110 or more. Authors should clarify and explain these unexpected behaviors.

According to the caption the cavity length has been changed with 1 μm steps. For each value of the cavity length 10 traces corresponding to different bunch trains are shown. For sake of simplicity traces corresponding to the same Dz are stacked and appears in the plot associated to Dz values different than the used one. Traces for $Dz=0$ are plotted with DZ ranging from -0.5 μm to 0.5 μm etc. If this is the case it may be recommended to put grids in the plot to isolate data with same Dz . As it is now, the plot suggests that Dz is varied continuously.

Around $Dz = -4\mu\text{m}$ it appears that only 2 traces are plotted and not 10 as mentioned in the caption.

The right panel is not clear. Caption suggests that each point should be the integral of one of the traces plotted on the left panel. If this is the case one expects to see 10 points for each Dz value (i.e. -4,-3,-2,-1,0,1...). The plot instead reports many more points suggesting again that Dz is varying continuously.

 - We understand the criticism on Fig. 3. It has been replaced by a map showing the downstream single shot spectrometer signal against bunch number and longitudinal detune. This has the advantage that the spectrometer (which had low attenuation for this dataset) is saturating at much higher intensities, and a round trip gain against Δz can be computed (on the right of the updated Fig.3). As for the former figure 3, per 1 μm bin the ten strongest samples are taken. However, unlike before, they have been averaged into a single line to

yield a more comprehensive picture and not give the false impression of a continuously varying Dz . The same was done for the old figure 3 with the diode signal, which has been shifted to Extended Data Figure 1.

5. Measurements of the cavity response indicate a significantly lower cold cavity round trip reflectivity than the theoretical one. Authors may comment on the required alignment efforts for reaching this value and if an improvement can be expected with some extra effort.
 - As suggested by the referee, non-proper alignment indeed has a strong impact on the cavity reflectivity, especially due to the comparably small size of the mirrors and the, hence, induced aperture. Yet, it is not expected that further efforts on the alignment will impact the cavity reflectivity, as the final alignment in the experiment has been reached by optimizing the ringdown efficiency (the round trip gain, however, could indeed profit from additional efforts alignment due to an increased overlap between X-rays and electron beam). Also, the alignment of the individual segments has been frequently rechecked using the scintillating screens by probing clearly defined setpoints. In order to clarify this in the text, following section has been altered in line 176: "In addition, imperfect alignment of the mechanics can lead to stronger losses [40]. However, this effect is suspected to be small, as the cavity was aligned by maximizing the ringdown signal. In simulation (see Methods), the ringdown is already reduced to 85 % by including the finite size of the mirrors and their measured imperfections."
6. In Figure 5 Authors show the measured FEL amplification over the bunch train for 3 different trains. Data show a clear amplification up to a maximum followed by a decay and few oscillations. The effect is associated to heat load effects on the cavity elements. Authors should comment if the electron beam jitter along the train (bunch timing, energy, centroid) may be also contributing to these oscillations.
 - We thank the referee for drawing our attention to this point in our argumentation. Actually, there are a few arguments against jitter being the source of the instability at around $10\mu\text{J}$. First, the comparably stable oscillations and long oscillation period over multiple bunches observed in the displayed sample shots contradict a shot to shot jitter. Second, the impact of jitter should also be strongly apparent in the gain curve before "saturation". Saturation of the gain process, lowering the FEL gain and, hence, the impact of jitter, is only expected at much stronger signal strengths at around 1mJ (following the Ming Xie approximation and simulation). Third, the oscillatory behaviour of the signal can be well reproduced in simulation when including the impact of thermal load. In order to better highlight this, following section has been added to line 254 of the manuscript: "Due to the long oscillation period over multiple bunches, this cannot be explained by shot-to-shot fluctuations in the beam quality. Also, saturation of the FEL process is expected at much higher pulse energies at around 1mJ [33, 41]. Compared with simulation (see the black dashed line) and as discussed in the literature [25, 33, 41, 44, 45], this can be attributed to heat load effects on diamond crystals, which affect the FEL process in two ways."
7. In line 337 the comment of the difference between optical resonators and CBFEL is not clear. Are Authors referring to FEL oscillator operating in the visible regime or to "standard" solid state or gas lasers?
 - The comment was referring to "standard" solid state or gas laser oscillators. To clarify this, the sentence (now line 354) has been altered to: "In contrast to laser-oscillators in the visible, ..."
8. Line 340 and following. More details on e-beam properties would be beneficial. Also, an indication of the measured gain-length in SASE mode for the used e-beam could be given as a reference of the beam quality.
 - The paragraph on the electron beam optimization has been altered to follow the suggestion of the referee and include more data on the actual electron beam parameters. Measured peak current, bunch length, as well as linear chirp are noted in the text. Also, the measured longitudinal phase space has been added to Extended Data Fig. 2. Further, the measured gain length (in SASE mode) has been noted.
9. Line 395: Clarify whether ΔE_{ph} is the same as ΔE
 - Both quantities are the same. Occurrences of ΔE have been changed to ΔE_{ph} to remain consistent. Also, there was a typo for the value of ΔE_{ph} , which has now been fixed from 0.8eV to 0.08eV .
10. Line 401: A bracket is not closed

- Fixed
11. Line 413: Does this imply that only one spectrometer pixel is used for the analysis? Please specify
 - There was a typo in the pixel width ΔE_{ph} , which has been fixed from from 0.8eV to 0.08 eV, making it 25 pixels for the integration.
 12. Line 692: ERSF is probably ESRF
 - That is correct, the typo has been fixed.
 13. Line 706: While in the mail text the term CBXFEL has been used here XFEL is used.
 - XFEL was changed to Cavity Based X-ray FEL to stay consistent with the rest of the manuscript.

Reponse to Referees:

We would like to again thank all three referees for their thorough review, many valuable suggestions and their recommendation for the manuscript to be featured in Nature.

Referee 3:

„Minor“ points raised:

1. In the introduction “Applying these concepts, several large-scale X-ray FEL (XFEL) user facilities were built [5–10], ...”

If authors want to specifically refer to hard x-ray facilities, reference [5] should be removed. If authors want to refer to only SASE facilities also SXFEL in Shanghai needs to be mentioned (<https://www.mdpi.com/2076-3417/12/1/176>, <https://lssfo1.cas.cn/en/facilities-view.jsp?id=ff80808151cd1d840151e718b8a31488>).

- In order to keep the number of total citations inside the Nature guidelines, we follow the suggestion by the referee and restrict the mentioning to hard X-ray facility, thus crossing the FLASH facility from the references.

2. Caption of fig. 2

- Caption reports that 25 pulses are plotted why the figure shows 30.

- The caption has been changed and now correctly reports that 30 pulses are plotted

- The comment on the double peak feature of some pulses is not clear. Why the subsequent electron should produce a signal at 10s or 100s of ns?

The relevant separation is between the minor peak and the next dominant peak (main signal peak). The separation of $\Delta t \approx 267$ ns fits well the time the a photon (or likewise the e-beam) needs to travel from the photodiode (where the e-beam causes the minor peak) to the downstream crystal and back to the photodiode (where the reflected and now directly incident beam causes the dominant peak). To better point this out in the manuscript, the following text has been added to the caption: “The $\Delta t \approx 267$ ns separation between the minor and the next dominant peak corresponds to the round-trip light path to the downstream crystal and back to the photodiode.”

3. Extended table

- It should specify if rms is used for jitter values.

- Has been added to the table to better point this out.

- Central chirp is missing μm^{-1}

- Has been fixed and now correctly shows the unit.

- Not clear what the 4 mm refer to.

- This was a typo. The entry has been deleted.

4. Simulations:

Authors report that "the undulator magnetic field strength was adjusted so that the round-trip gain matches the one measured in the experiments" Is this meaning that in the simulation the undulator resonance condition has been changed to allow reproducing the experimental results? What was the offset required? From the spectra in Fig.4 one would say that the undulator resonance is ok within 5×10^{-4} .

The undulator parameter was used in simulation as a simple „free“ parameter to include unknown, and more involved parameters especially with regard to the electron beam, such as electron beam tilt or electron beam slice emittance at the undulators. It has been adjusted by 2.5×10^{-4} to better match the results with special regard to intensity, and does not perfectly correspond to the one used in the experiment. In order to better point this out in the manuscript, the line 591 (line 469 old manuscript): "... campaign. The undulator magnetic field strength was treated as an adjustable parameter, enabling a slight shift of the peak of the SASE spectral envelope away from the reflection curve by a fraction of the SASE bandwidth. This adjustment reduced the seeding gain to replicate the experimentally observed round-trip gain and to account for additional, unresolved sources of gain reduction, such as imperfections in the electron beam distribution."

Authors report "by adjusting the Undulator gain, the strength of the thermal oscillations could be influenced ..." If this is the case, the same should be for electron beam energy jitter, but being of 1×10^{-4} maybe this is a smaller effect.

- Increasing the electron beam energy jitter would certainly impact average FEL gain, and, thus, also alter the strength of the thermal oscillations. However, as increasing the jitter also direct has an influence on the shot-to-shot jitter, its effect is more involved than the „static“ change of the undulator parameter and has not been studied in detail.